# Comprehensive increase in $CO_2$ release by drying-rewetting cycles among Japanese forests and pastureland soils and exploring predictors of increasing magnitude

Yuri Suzuki[1,2], Syuntaro Hiradate[3], Jun Koarashi[4], Mariko Atarashi-Andoh[4], Takumi Yomogida[4], Yuki Kanda[1], and Hirohiko Nagano[5]

[1]Faculty of Agriculture, Niigata University, Niigata 950-2181, Japan
[2]Graduate School of Environmental Science, Niigata University, Niigata 950-2181, Japan
[3]Graduate school of Agriculture, Kyushu University, Fukuoka 819-0395, Japan
[4]Nuclear Science and Engineering Center, Japan Atomic Energy Agency, Tokai, Ibaraki 319-1195, Japan
[5]Institute of Science and Technology, Niigata University, Niigata 950-2181, Japan

**Correspondence:** Hirohiko Nagano (hnagano@agr.niigata-u.ac.jp)

**Abstract.** It is still difficult to precisely quantify and predict the effects of drying-rewetting cycles (DWCs) on soil carbon dioxide ($CO_2$) release due to the paucity of studies using constant moisture conditions equivalent to the mean water content during DWC incubation. The present study was performed to evaluate overall trends in the effects of DWCs on $CO_2$ release and to explore environmental and soil predictors for variations in the effect size in 10 Japanese forests and pastureland soils variously affected by volcanic ash during their pedogenesis. Over an 84-day incubation period including three DWCs, $CO_2$ release was 1.3- to 3.7-fold greater than under continuous constant moisture conditions ($p < 0.05$) with the same mean water content as in the DWC incubations. Analysis of the relations between this increasing magnitude of $CO_2$ release by DWCs ($IF_{CO2}$) and various environmental and soil properties revealed significant positive correlations between $IF_{CO2}$ and soil organo-metal complex contents ($p < 0.05$), especially pyrophosphate extractable aluminum (Alp) content ($r = 0.74$). Molar ratios of soil total carbon (C) and pyrophosphate-extractable C (Cp) to Alp contents and soil carbon content-specific $CO_2$ release rate under continuous constant moisture conditions ($qCO_2\_soc$) were also correlated with $IF_{CO2}$ ($p < 0.05$). The covariations among Alp, total C, and Cp to Alp molar ratios and $qCO_2\_soc$ suggested Alp as the primary predictor of $IF_{CO2}$. Additionally, soil microbial biomass C and nitrogen (N) levels were significantly lower in DWCs than under continuous constant moisture conditions, whereas there was no significant relation between the microbial biomass decrease and $IF_{CO2}$. The present study showed a comprehensive increase in soil $CO_2$ release by DWC in Japanese forests and pastureland soils, suggesting that Alp is a predictor of the effect size likely due to vulnerability of organo-Al complexes to DWC.

# 1 Introduction

There is accumulating evidence of climate change-induced alterations in global water cycling (IPCC, 2021; Allan et al., 2020;
Dai, 2012; Donat et al., 2016; Pfleiderer et al., 2019). Of the consequent water regime changes, decreasing precipitation
frequency (e.g., number of rainy days) and increasing intensity (e.g., number of heavy rainy days) are becoming more frequent
(Dai, 2012; Donat et al., 2016), although decadal trends in annual precipitation levels are not significant over global scales
(IPCC, 2021). For example, the annual number of precipitation days in Japan has decreased by 15% during the past 120 years,
whereas the annual number of heavy precipitation days (more than 100 mm in a day) has increased by 26% (Ministry of
Education, Science, and Technology (MEXT) and Japanese Meteorological Agency (JMA), 2020). There is a non-significant
trend in annual precipitation level during the same period (MEXT and JAM, 2020). This changing pattern in precipitation
is often observed in the temperate region of the northern hemisphere (IPCC, 2021) and is related to increased fluctuation of
soil water environments, especially DWCs, and consequent alterations in ecosystem functions (Borken and Matzner, 2009; Jin
et al., 2023; Zhang et al., 2020, 2023).

Carbon dioxide ($CO_2$) release from soil is an ecosystem process that is sensitive to DWCs (Birch, 1958; Borken and Matzner,
2009; Lee et al., 2002; Nagano et al., 2019; Unger et al., 2010, 2012; Zhang et al., 2020, 2023) and has substantial feedback
potential to the ongoing climate change due to its magnitude reaching as much as seven times greater than anthropogenic $CO_2$
emission on a global scale (Bond-Lamberty and Thomson, 2010; Friedlingstein et al., 2020). The effects of DWCs on soil $CO_2$
release were first shown by Birch (1958) as the marked increase in soil organic matter (SOM) decomposition and $CO_2$ release
after the rapid rewetting of dried soil, and has since been the subject of intensive investigation (Borken and Matzner, 2009;
Kpemoua et al., 2023; Lee et al., 2002, 2004; Miller et al., 2005; Nagano et al., 2019; Unger et al., 2010, 2012; Xiang et al.,
2008), including meta-analyses (Kim et al., 2012; Jin et al., 2023; Zhang et al., 2020). However, it is still difficult to precisely
quantify and predict the effects of DWCs on soil $CO_2$ release.

The significant uncertainties in the effects of DWCs on soil $CO_2$ release include the inconsistent trends and sizes of effects
likely due to the paucity of studies using constant moisture conditions equivalent to the mean water content during DWC
incubation (Kpemoua et al., 2023; Zhang et al., 2020, 2023). According to a meta-analysis by Zhang et al. (2020) using 208
data from 34 sites in 29 reports, the effects of DWCs vary according to soil and water contents in continuous constant moisture
conditions. Especially, changes in $CO_2$ release rate associated with DWCs ranged from -4% to +19% with an average of +4% in
comparison with the medium level of constant moisture content, which should be the same with the mean water content during
DWC incubation, whereas only 9 of 38 data representing $CO_2$ release rates were measured for such medium level of constant
moisture conditions (Zhang et al., 2020). Another 29 data were calculated as the average of two $CO_2$ release rates at the wettest
and driest water contents of constant moisture conditions, which should be the same with the maximum and minimum water
contents in DWC treatment, respectively (Zhang et al., 2020). In the experiment using three Alfisols from Chinese long-term
experimental field studies, Zhang et al. (2023) showed similar or somewhat lower $CO_2$ release in the DWC compared with
the constant moisture conditions with the same mean water content for the DWC incubation. Using two Luvisols from French
long-term field experiment sites, Kpemoua et al. (2023) also showed similar features of changes in $CO_2$ release associated with

DWCs, indicating the need for further comparison of $CO_2$ release between DWCs and constant moisture conditions with the same mean water content. In contrast to these studies, Nagano et al. (2019) found a 49% increase in $CO_2$ release rate associated with DWCs in an Andisol collected from a Japanese forest. This increase was more than double that of another non-volcanic ash soil from the same forest. Thus, there are substantial variations in trends of effects of DWCs in comparison with constant moisture conditions having the same mean water content during incubation, remaining knowledge gaps about environmental and soil predictors for variations in effect sizes. There are proposed roughly three mechanisms for $CO_2$ release increase by DWCs (Schimel, 2018; Barnard et al., 2020): (i) increase in available carbon source via the releases of cellular metabolites from microbial cells destroyed by rewetting after the strong drought, (ii) increase in available carbon source by the releases of carbon from macroaggregates destroyed by repeated DWC, and (iii) changes in the microbial communities in response to transient moisture conditions. Nevertheless, there are still substantial knowledge gaps for critical mechanisms or the relative importance of those mechanisms among multiple soils.

We perform the present study to evaluate overall trends in the effect of DWCs on soil $CO_2$ release and to explore the predictors of variations in its effect size among 10 Japanese forests and pastureland soils. These soils are variously affected by volcanic ash during their pedogenesis and, therefore, include several Andisols, which are known to have a high SOM storage capacity (Morisada et al., 2004), likely due to the protection of SOM from microbial decomposition by abundant reactive minerals and metals in these soils (Asano and Wagai, 2014; Imaya et al., 2007; Shirato et al., 2004). Reactive minerals and metals that contribute to the protection of SOM are iron (Fe) and aluminum (Al), constituting short-range-order minerals and organo-metal complexes (Asano and Wagai, 2014; Rasmussen et al., 2018; Shirato et al., 2004; Wagai et al., 2018). Although global coverage of Andisols is about 1% (FAO/IIASA/ISRIC/ISS-CAS/JRC, 2009), determination of the responses of carbon cycling in Andisols to DWCs will help in understanding the responses of non-volcanic ash soils, because reactive minerals and metals are also essential in high carbon stocks of those soils (Rasmussen et al., 2018; Hall and Thompson, 2022) and may be sensitive to climate and land use management, including water regime (Georgiou et al., 2022; Kramer and Chadwick, 2018).

## 2 Materials and methods

### 2.1 Site description and soil sampling

We collected 10 soil samples from depths of 0–5 cm or 0–10 cm in six forests and a pastureland located in Niigata (six soils from four forests), Ibaraki (two soils from a forest) (Nagano et al., 2019), and Oita (two soils from a forest and a pastureland) (Wijesinghe et al., 2021) prefectures in Japan. Figure 1 and Table 1 present the locations and site characteristics, i.e., elevation, mean annual temperature (MAT), mean annual precipitation (MAP), potential evapotranspiration (PET), and net primary production (NPP). Briefly, all of the investigated sites have a humid temperate climate with MAT of 9.1–10.8°C and MAP of 1474–2930 mm. All of the forests are dominated by beech (*Fagus crenata* and *Fagus japonica*) and oak (*Quercus serrata*), except for Oita forest which is a deciduous/evergreen mixed forest dominated by Siebolds maple (*Acer sieboldianum*), Japanese snowbell (*Styrax japonicus*), and Japanese holly (*Ilex crenata*). The pastureland in Oita is dominated by Japanese lawn grass

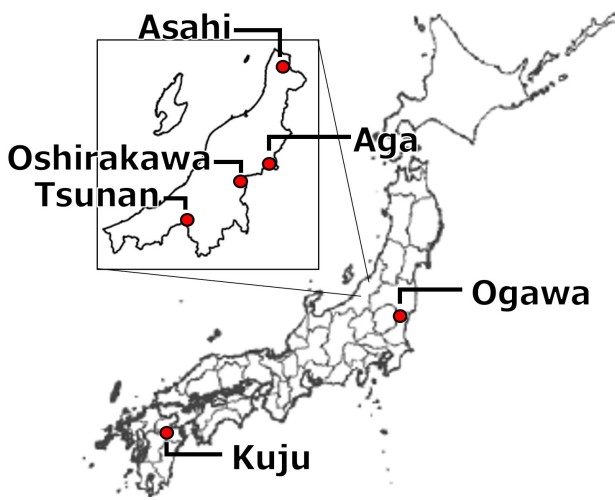

**Figure 1.** Locations of Japanese forests and pastureland where investigated soils were collected.

**Table 1.** Environmental properties of Japanese forests and pastureland where investigated soils were collected[a]

| Prefecture | Soil | Latitude °N | Longitude °E | Elevation m | MAT °C | MAP mm | Annual PET mm | MAP-PET mm | Annual NPP g C m$^{-2}$ |
|---|---|---|---|---|---|---|---|---|---|
| Niigata | Aga1, Aga2 | 37.55 | 139.51 | 474 | 10.6 | 1714 | 747 | 967 | 633 |
| | Tsunan1, Tsunan2 | 37.04 | 138.60 | 716 | 10.8 | 1899 | 795 | 1104 | 628 |
| | Asahi1 | 38.38 | 139.70 | 655 | 9.2 | 1887 | 697 | 1190 | 605 |
| | Oshirakawa1 | 37.35 | 139.16 | 627 | 9.1 | 1963 | 668 | 1295 | 563 |
| Ibaraki | Ogawa13, Ogawa 14 | 36.93 | 140.59 | 643 | 10.7 | 1474 | 764 | 710 | 785 |
| Oita | Kuju_pasture, Kuju_forest | 33.06 | 131.23 | 841 | 10.8 | 2930 | 786 | 2144 | 819 |

[a] MAT, MAP, and PET were obtained as averages for 1981–2020 in a global data set of climate and climatic water balance (i.e., TerraClimate) by Abatzoglou et al. (2018). NPP data were obtained as averages for 2001–2020 in the global distribution of NPP estimated from MODIS observation products (i.e., MOD17A3HGF) by Running and Zhao. (2021). MAT, mean annual temperature; MAP, mean annual precipitation; PET, potential evapotranspiration; NPP, net primary production.

(*Zoysia japonica*), dwarf fountain grass (*Pennisetum alopecuroides*), cranesbill (*Geranium thunbergii*), white clover (*Trifolium repens*), and Indian strawberry (*Potentilla indica*).

Soil sampling was conducted in the snow-free season (April to October) of 2021. We also collected soil samples from layers below the target depth (i.e., 0–5 cm or 0–10 cm depth) down to 50 cm as the maximum depth to examine whether the soil could be classified as Andisol. According to the USDA (United States Department of Agriculture) Soil Taxonomy criteria (USDA Soil Survey Staff, 2022), soils with 60% or more of the thickness containing more than 20 mg of acid oxalate-extractable Al (Alo) plus 1/2 Fe (Feo) per 1 g soil within a depth of 0–60 cm are classified as Andisols. According to these criteria, 4 of the 10 soils were Andisols (one each from Niigata and Ibaraki, and two from Oita) with non-allophanic properties determined by a

high ratio of pyrophosphate-extractable Al (Alp) content to Alo content (Alp/Alo > 0.5; USDA Soil Survey Staff (2022); Fig. S1). Collected soil samples were transferred to the laboratory and stored at 4°C before further analysis. Before analysis, soils were gently passed through a 4-mm sieve to remove gravel and plant tissue. Fine roots in the sieved samples were removed with tweezers. In our study, we considered that the soil water content at the soil sampling reflected the ability of soil to hold the water and thus the usual water contents in the field because the soil water content showed significantly positive correlations with water holding capacity (WHC) ($r = 0.87$, $p < 0.01$). Therefore, $CO_2$ release rate for constant moisture conditions in the present study should represent the release rate under the usual field moisture conditions of each soil.

## 2.2 Soil analysis

The soil properties analyzed were pH($H_2O$), electrical conductivity, water content, water-holding capacity (WHC), total carbon (C) and nitrogen (N) contents, particle size distributions as relative compositions of clay, silt, and sand-sized particles, and selectively dissolved Al and Fe contents (Tables 2 – 4). The pH($H_2O$) was measured in soil and water mixtures consisting of 1 g of soil and 2.5 mL of water. For measurement of electrical conductivity, 5 mL of water was added to 1 g of soil. Water content was measured by determining the difference in soil weight before and after drying at 105°C for 24 h. WHC was measured as the difference in soil weight before and after water saturation referring to the Hilgard method (Mabuhay et al., 2003; Ahn et al., 2008). Here, water contents when soil is completely saturated in the Hilgard method should equal zero pF value (0 kPa) as soil water potential. Soil total C and N contents were measured for air-dried and well-ground soil samples using an elemental analyzer (vario PYRO cube; Elementar, Manchester, UK). Particle size distributions were determined using the Stokes' law-based sedimentation method (Miller et al., 1988; Nakai, 1997) using soil mineral particles after removing organic matter by hydrogen peroxide solution digestion. Selectively dissolved metals such as reactive Al and Fe were measured according to the procedure described previously in Nagano et al. (2023). Briefly, the contents of Al and Fe extractable with 2.0 M acid ammonium oxalate (i.e., Alo and Feo, respectively) were measured as contents of organo-metal complexes and short-range-order minerals, while Al and Fe extractable with 0.1 M sodium pyrophosphate solution (i.e., Alp and Fep, respectively) were measured as contents of organo-metal complexes (Asano and Wagai, 2014; Takahashi and Dahlgren, 2016; Wagai et al., 2018). The difference between acid oxalate- and pyrophosphate-extractable metals (i.e., Alo-p and Feo-p) represented the contents of short-range-order minerals (Courchesne and Turmel, 2008). Contents of Al and Fe in the solution were measured with an inductively coupled plasmaoptical emission spectrometer (ICP-OES) (5110; Agilent Technologies, Santa Clara, CA, USA). Carbon concentrations in pyrophosphate-extracted solution (Cp) were also measured with a total organic carbon (TOC) analyzer (TOC-L; Shimadzu, Kyoto, Japan). For soils from Kuju forest and grassland and two Ogawa forests, we also measured C concentration in free light density fraction (fLF) (Leuthold et al., 2023) obtained by the density fractionation method using sodium polytungstate solution having 2.0 g cm$^{-3}$ (Koarashi et al., 2012), in order to evaluate possible effects of labile C abundance in fLF on $CO_2$ release increase by DWCs.

For the soils after incubation (see Section 2.3), soil microbial biomass C and N were measured by the chloroform fumigation-extraction method (Vance et al., 1987). Organic C and total N concentrations in 0.5 M potassium sulfate solution used for

**Table 2.** Basic properties of 10 Japanese forests and pastureland soils collected from depths of 0–5 or 0–10 cm

| Soil | Soil type[a] | pH(H$_2$O) | EC | Water content | WHC | Particle size-distribution | | |
|---|---|---|---|---|---|---|---|---|
| | | | | | | Sand | Silt | Clay |
| | | | $\mu$S cm$^{-1}$ | g water g$^{-1}$ soil | g water g$^{-1}$ soil | % | | |
| Aga1 | Inceptisols or Entisols | 4.36 | 23 | 0.77 | 0.80 | 27 | 30 | 43 |
| Aga2 | Inceptisols or Entisols | 4.39 | 27 | 1.07 | 1.12 | 17 | 59 | 23 |
| Tsunan1 | Inceptisols (Brown forest soils) | 4.72 | 17 | 0.78 | 1.06 | 47 | 44 | 9 |
| Tsunan2 | Inceptisols (Brown forest soils) | 4.08 | 39 | 1.06 | 1.32 | 34 | 48 | 18 |
| Asahi1 | Inceptisols (Brown forest soils) | 4.07 | 39 | 1.70 | 2.14 | 29 | 41 | 30 |
| Oshirakawa1 | Andisols | 4.66 | 24 | 1.07 | 1.26 | 38 | 45 | 17 |
| Ogawa13 | Inceptisols (Brown forest soils) | 5.29 | 18 | 0.46 | 1.00 | 66 | 25 | 8 |
| Ogawa14 | Andisols | 5.30 | 27 | 1.14 | 1.68 | 68 | 23 | 9 |
| Kuju_pasture | Andisols | 5.35 | 133 | 1.32 | 1.43 | 72 | 22 | 5 |
| Kuju_forest | Andisols | 3.80 | 168 | 1.49 | 1.58 | 70 | 25 | 4 |

[a] Andisols were determined according to the USDA Soil Taxonomy criteria (USDA Soil Survey Staff, 2022) based on acid oxalate-extractable Al plus 1/2 Fe contents (see text for details). All Andisols were non-allophanic. Other soil types were determined using a Japanese soil digital map, i.e., Japan Soil Inventory NARO (2023).

extraction of fumigated and nonfumigated soils were measured using another TOC analyzer (TOC-L; Shimadzu) equipped with a total nitrogen (TN) unit (TNM-L, Shimadzu).

### 2.3 Incubation experiment and soil CO$_2$ release rate measurement

Soils were incubated aerobically at 20°C for 84 days including three DWCs (i.e., 28 days per cycle). Simultaneously, soils were incubated in the same manner but without DWCs, during which water content of the soils was maintained at a constant level equivalent to the mean water content for the DWC treatment (Fig. 2). A pre-incubation was conducted at the constant water content for 7 days prior to the 84-day incubation. A post-incubation was also conducted at the constant water content for 28 days after the 84-day incubation to evaluate the remaining effect of DWCs on soil CO$_2$ release. We consider that the incubated soils have been aerobic even after the rewetting to increase the water content by twice the WHC, because the CO$_2$ concentrations in our experiment never overwhelmed 1%, thus the oxygen concentrations in the incubation jar have likely never

**Table 3.** Soil carbon (C) and nitrogen (N) properties

| Soil | Total C | Total N | Total C/N | C in free light density fraction (fLF) |
|---|---|---|---|---|
| | % | % | | % to total C |
| Aga1 | 8.8 | 0.56 | 15.7 | No data |
| Aga2 | 14.3 | 0.79 | 18.2 | No data |
| Tsunan1 | 12.4 | 0.63 | 19.8 | No data |
| Tsunan2 | 15.3 | 0.98 | 15.6 | No data |
| Asahi1 | 23.7 | 1.41 | 16.9 | No data |
| Oshirakawa1 | 15.1 | 0.76 | 19.8 | No data |
| Ogawa13 | 6.8 | 0.31 | 22.2 | 10.0 |
| Ogawa14 | 16.1 | 0.86 | 18.6 | 10.0 |
| Kuju_pasture | 21.8 | 1.37 | 15.9 | 14.0 |
| Kuju_forest | 22.3 | 1.18 | 18.9 | 5.9 |

**Table 4.** Selectively dissolved minerals and associated carbon contents in soils

| Soil | Acid oxalate-extractable metals | | | Pyrophosphate-extractable metals | | | Acid oxalate – Pyrophosphate | | | Alp/Alo | Pyrophosphate-extractable C |
|---|---|---|---|---|---|---|---|---|---|---|---|
| | Al | Fe | Al+0.5Fe | Al | Fe | Al+0.5Fe | Al | Fe | Al+0.5Fe | | |
| | $mg\ g^{-1}$ dry soil | | | $mg\ g^{-1}$ dry soil | | | $mg\ g^{-1}$ dry soil | | | | $mg\ g^{-1}$ dry soil |
| Aga1 | 3.3 | 7.9 | 7.2 | 2.8 | 5.0 | 5.3 | 0.5 | 2.8 | 2.0 | 0.84 | 21.0 |
| Aga2 | 6.9 | 10.6 | 12.2 | 6.8 | 10.5 | 12.0 | 0.1 | 0.2 | 0.2 | 0.98 | 42.9 |
| Tsunan1 | 8.9 | 13.8 | 15.8 | 8.5 | 11.3 | 14.2 | 0.4 | 2.5 | 1.6 | 0.96 | 39.9 |
| Tsunan2 | 6.9 | 8.5 | 11.1 | 7.1 | 7.2 | 10.6 | -0.2 | 1.4 | 0.5 | 1.03 | 47.4 |
| Asahi1 | 4.8 | 10.2 | 9.9 | 4.7 | 9.2 | 9.3 | 0.1 | 1.0 | 0.6 | 0.98 | 45.4 |
| Oshirakawa1 | 9.5 | 12.4 | 15.7 | 9.1 | 11.0 | 14.6 | 0.4 | 1.4 | 1.2 | 0.95 | 43.0 |
| Ogawa13 | 9.2 | 8.7 | 13.5 | 6.9 | 6.5 | 10.2 | 2.3 | 2.2 | 3.4 | 0.75 | 19.4 |
| Ogawa14 | 24.9 | 16.6 | 33.2 | 23.2 | 12.7 | 29.6 | 1.6 | 3.9 | 3.6 | 0.93 | 59.1 |
| Kuju_pasture | 22.0 | 20.3 | 32.2 | 19.0 | 14.1 | 26.0 | 3.1 | 6.3 | 6.2 | 0.86 | 96.7 |
| Kuju_forest | 19.5 | 19.1 | 29.0 | 19.0 | 16.2 | 27.1 | 0.4 | 2.9 | 2.0 | 0.98 | 109.8 |

decreased below 19% or lower. Also, a sufficiently large volume of our incubation jar (1.0L) against contained soil amounts (i.e., 5.31-10.63 g) and added water contents in the rewetting (i.e., ca. 6 to 7 mL) support the state of aerobic condition during the incubation.

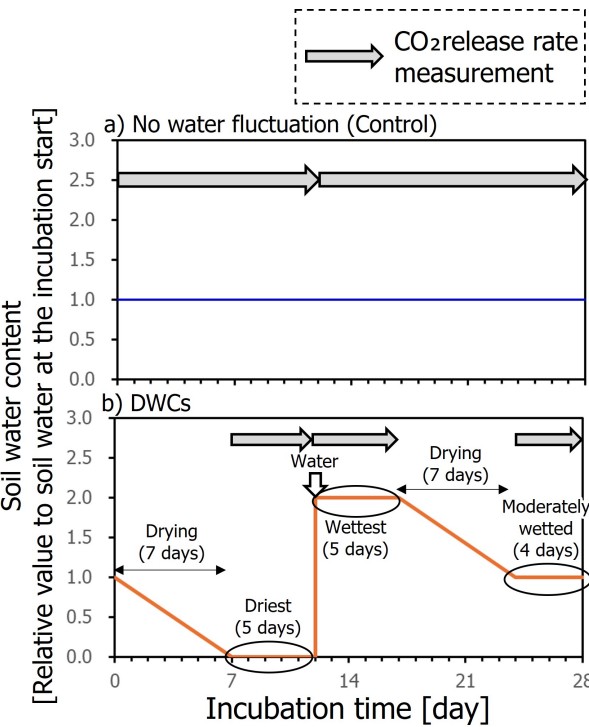

**Figure 2.** Schematic time courses of soil moisture during incubation with constant moisture content (a) and DWCs (b) with measurement period of $CO_2$ release rate.

Mason jars (1.0 L volume; Ball, Buffalo, NY, USA) with lids equipped with tube fitting systems for gas sample collection (Koarashi et al., 2012; Nagano et al., 2019) were used as incubation jars. Small vials (300 mL, SM sample glass vial; Sansyo, Tokyo, Japan) containing 5.31–10.63 g of soil sample depending on water content were placed in Mason jars. Under the constant water treatment, we surrounded the small vial with 20 mL of water within the incubation jar to prevent the soil from drying. For each DWC, Day 1 to Day 7 and Day 18 to Day 24 were drying stages (Fig. 2), during which the soils were incubated with silica gel (20 g jar$^{-1}$), which lowered the water content to < 5% WHC by Day 7. Day 8 to Day 12, Day 13 to Day 17, and Day 25 to Day 28 were the driest, wettest, and moderately wet stages, respectively, and $CO_2$ release rates in these three stages were measured using a gas chromatograph equipped with a thermal conductivity detector (GC-14B; Shimadzu). The $CO_2$ release rates were also measured in the pre- and post-incubation periods. At the beginning of the wettest stage, soils were rapidly rewetted with distilled water to double the soil water content from the initial status of DWC incubation. For $CO_2$ release rate measurements, at the start of each stage, the headspace of the incubation jar was flushed with $CO_2$-free air for 15 min at a rate of 0.5 L min$^{-1}$ and the jar was closed. At the end of each stage, 15 mL of gas sample was collected from the jar using a 20-mL plastic syringe (Terumo, Kyoto, Japan) and stored in a pre-evacuated 5-mL glass vial (SVG-5; Nichiden Rika, Osaka, Japan). Then, the $CO_2$ release rate was determined from the increase in $CO_2$ concentration during this period. After gas sampling, the jars were flushed with $CO_2$-free air and closed for the next incubation stage. For 84 days in the incubations

with the constant moisture condition as controls, the $CO_2$ release rates were measured for Day 1 to Day 12, Day 13 to Day 28, Day 29 to Day 40, Day 41 to Day 56, Day 57 to Day 68, and Day 69 to Day 84, in addition to the pre- and post-incubation periods. Soil water contents during the incubation were measured periodically and maintained by adding water to ensure the same mean water content between the two treatments. Even in the drying stage under the DWC treatment for Day 1 to Day 7 and Day 18 to Day 24, we conducted measurements of soil water content once to twice. The measurements were performed by weighing those soils. Based on these data, we confirmed that the mean soil water content during DWC incubation was equal to that during constant moisture incubation. All incubations were conducted with three replicates for each treatment and soil.

## 2.4 Data processing and statistical analysis

The $CO_2$ release rates were compared between the DWC and constant water content treatments. For the DWC treatment, the $CO_2$ release rates in the drying stages (i.e., Day 1 to Day 7 and Day 18 to Day 24) could not be measured and therefore had to be estimated to evaluate the mean $CO_2$ release rates for the individual cycle and total of three cycles. The rates in the drying stages were estimated as the mean values of the $CO_2$ release rates measured before and after the period of interest, in the same manner as described previously (Nagano et al., 2019). Then, the effect size of the DWCs on $CO_2$ release (defined here as the increase factor, $IF_{CO2}$) was quantified as the ratio of $CO_2$ release rate under the DWC condition to that under the constant water content condition (Nagano et al., 2019; Zhang et al., 2020). Here, doubling of $CO_2$ release by DWC resulted in $IF_{CO2}$ of 2, while halving resulted in $IF_{CO2}$ of 0.5.

The pairwise *t* test was applied to examine the statistical significance of differences in $CO_2$ release rates between the DWC and constant water content treatments with adjustment of site-by-site variations in the metrics. Differences in soil microbial biomass C and N between the two treatments were also evaluated with the pairwise *t* test. To explore predictors explaining the variation in $IF_{CO2}$ among soils, relations between $IF_{CO2}$ and environmental and soil properties were visualized on scatter plots and evaluated by linear correlation analysis. All statistical analyses were conducted with R 4.1.1 (R Core Team, Vienna, Austria), and $p < 0.05$ was taken to indicate statistical significance.

## 3 Results

### 3.1 Quantifying the effect of DWCs on soil $CO_2$ release

The $CO_2$ release rates under DWC conditions showed large fluctuations for all soils along with fluctuations in soil water content (Fig. 3). The $CO_2$ release rates in the driest stages (3.5%–18.2% of WHC depending on the soil) were 3.0–41.5 $\mu$g C g$^{-1}$ dry soil day$^{-1}$. In contrast, $CO_2$ release rates in the wettest stages (90.8%–201.1% of WHC) reached 18.8–194.1 $\mu$g C g$^{-1}$ dry soil day$^{-1}$. In the moderately wet stages, $CO_2$ release rates were 8.4–63.9 $\mu$g C g$^{-1}$ dry soil day$^{-1}$. For the constant water content treatment, $CO_2$ release rates observed for each soil showed little fluctuation during incubation although the rates varied from 7.3 to 69.9 $\mu$g C g$^{-1}$ dry soil day$^{-1}$ depending on the soil.

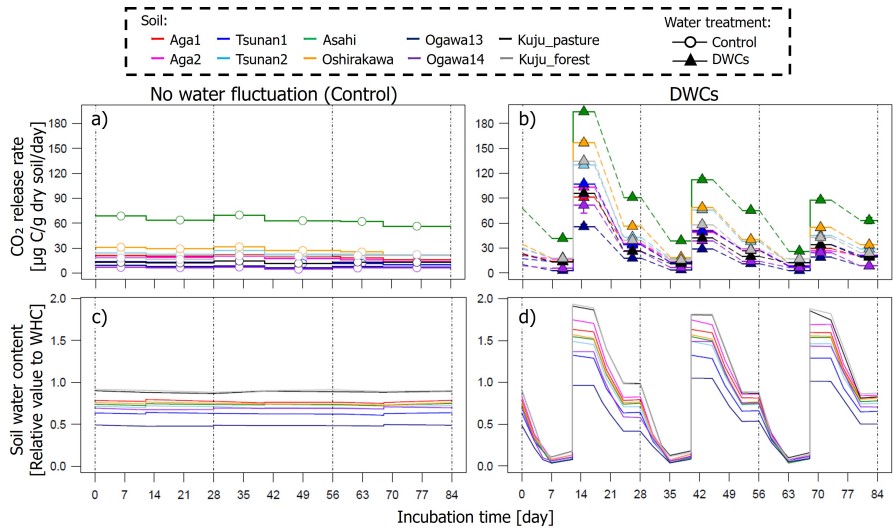

**Figure 3.** Time courses of soil $CO_2$ release rates (a and b) and water contents (c and d) during 84-day incubation under constant water content (a and c) and three DWC conditions (b and d).

For all soils, the observed large fluctuations in $CO_2$ release under DWC treatment resulted in a greater $CO_2$ release rate than under constant water content treatment, although the $IF_{CO2}$ values varied among the soils (Fig. 4). At the wettest stage in the first DWC, the $CO_2$ release rates increased by 47.2–127.7 $\mu$g C g$^{-1}$ dry soil day$^{-1}$ compared with those under the constant water content conditions, resulting in $IF_{CO2}$ values of 2.9–12.2. For the whole of the first cycle (i.e., the first 28 days), the $CO_2$ release rates increased by 15.4–43.4 $\mu$g C g$^{-1}$ dry soil day$^{-1}$ under the DWCs compared with the constant water content conditions, resulting in $IF_{CO2}$ values of 1.6–5.2. For the whole incubation period (84 days) including three DWCs, $IF_{CO2}$ values were 1.3–3.7, with an increase in $CO_2$ release rate by 7.4–23.8 $\mu$g C g$^{-1}$ dry soil day$^{-1}$ by DWCs. These increases in $CO_2$ release by DWCs were observed in all cycles during the 84-day incubation period, whereas no increase was observed in the 28-day post-incubation period after the three DWCs (Fig. S2). Considering these results, we focused on the $IF_{CO2}$ variations obtained for the whole incubation period, including three DWCs.

### 3.2 Exploring predictors of the effect sizes on the increase in $CO_2$ release

Among the environmental and soil physiochemical properties, reactive mineral and metal contents (especially, Alo+0.5Feo, Alo, Feo, Alp+0.5Fep, and Alp) in soils showed significant positive correlations with $IF_{CO2}$ ($p < 0.05$; Table 5). In particular, Alp content appeared to be a key predictor of the variation in $IF_{CO2}$ among the soils, given that Alp accounted for most (73%–99%) of Alo (Table 4) and showed a higher correlation coefficient than others (Table 5). Scatterplots for Alp content and $IF_{CO2}$ values are presented in Fig. 5. In addition, the molar ratios of soil total C and Cp contents to Alp contents showed significant negative correlations to $IF_{CO2}$ ($p < 0.05$; Table 5, Fig. 5). Here, it should be noted that $IF_{CO2}$ had no significant relationship with soil water content at soil sampling and WHC, suggesting the variation in $IF_{CO2}$ among soils resulting from

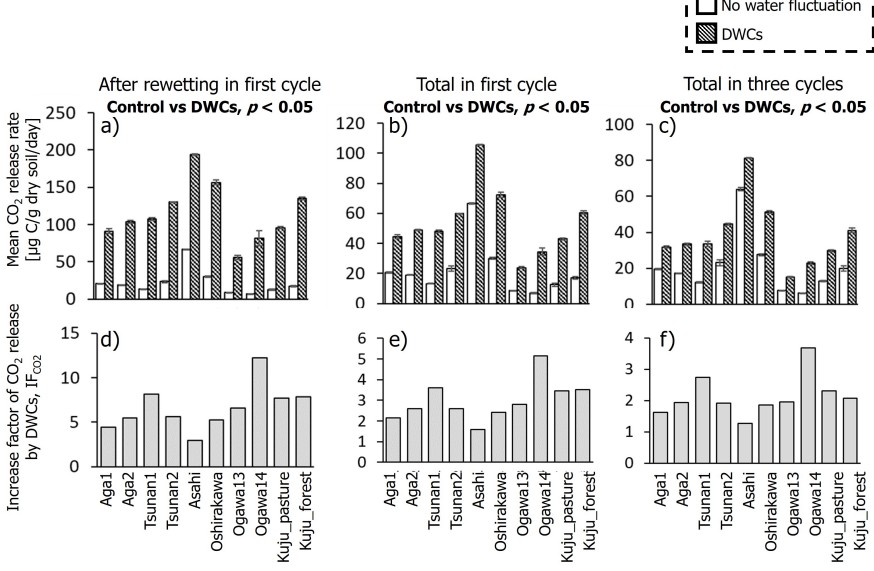

**Figure 4.** Comparisons of mean $CO_2$ release rates after rewetting in the first cycle, the whole of the first cycle, and for the total of three cycles between DWC and constant water content conditions (a-c), and the factor of increase in $CO_2$ release by DWCs ($IF_{CO2}$) for individual periods (d-f). Statistically significant differences ($p < 0.05$, pairwise $t$ test) in $CO_2$ release rate between the two treatments are presented.

other than such hydrogenic properties of soils. The SOM quality, such as C/N ratios of total bulk soils and $K_2SO_4$ extractable fractions, also had no signifcant relationship with $IF_{CO2}$. The insignificant relations of SOM quality to $IF_{CO2}$ were also supported by the facts that Kuju forest and grassland soils having more than doubling differences in fLF abundance showed only 12% differences in $IF_{CO2}$, and two Ogawa forest soils having almost identical fLF abundances showed more than doubling

differences in $IF_{CO2}$. The amounts of clay and sand-sized particles showed significant correlations with $IF_{CO2}$ after rewetting in the first cycle ($p < 0.05$, $r$ = -0.66 for clay and 0.71 for sand particles). However, those correlations between the particle contents and $IF_{CO2}$ were insignificant for $IF_{CO2}$ for a total of three cycles. Thus, in the present study, reactive mineral and metal content, especially Alp content in soils, was the significant predictor for $IF_{CO2}$ variation among soils, rather than soil water environments, SOM quality, and particle size distributions.

Among the additional soil properties related to soil microbial activity and abundance, the soil C content-specific $CO_2$ release rate under constant moisture conditions (defined as the $CO_2$ release rate per unit of C in soil, $qCO_2\_soc$, in $\mu g\ CO^2$-C mg$^{-1}$ soil-C day$^{-1}$) showed significant negative correlations with $IF_{CO2}$ values in all incubation stages ($p < 0.01$; Table 6, Fig. 6). Here $qCO_2\_soc$ should be an index for microbial availability of carbon substrate normalized by total C contents in soils. Thus, using $qCO_2\_soc$, we can consider whether the microbially available carbon substrate in interested soil should be much more

than in other soils. Microbial biomass N also showed significantly negative correlations with $IF_{CO2}$ values after rewetting in the first cycle and in the whole of the first cycle ($p < 0.05$) but not in the whole incubation period including three cycles ($p$ = 0.12).

**Table 5.** Pearsons correlation coefficients between $IF_{CO2}$ and environmental and soil properties

| Environmental or soil property | After rewetting in first cycle | Total in first cycle | Total in three cycles |
|---|---|---|---|
| Elevation | 0.36 | 0.37 | 0.22 |
| MAT | 0.54 | 0.55 | 0.46 |
| MAP | 0.05 | 0.08 | -0.11 |
| MAP-PET | 0.01 | 0.04 | -0.14 |
| Water content at sampling | -0.11 | -0.07 | -0.15 |
| WHC | 0.06 | 0.08 | 0.04 |
| pH($H_2O$) | 0.54 | 0.51 | 0.57 |
| Electronic conductivity | 0.19 | 0.21 | -0.01 |
| Total C | -0.03 | 0.01 | -0.11 |
| Total N | -0.07 | -0.03 | -0.12 |
| C/N ratio | 0.27 | 0.23 | 0.22 |
| Sand | **0.72*** | **0.70*** | 0.56 |
| Silt | -0.50 | -0.48 | -0.35 |
| Clay | **-0.69*** | **-0.68*** | -0.57 |
| Alo+0.5Feo | **0.81**** | **0.82**** | **0.70*** |
| Alo | **0.84**** | **0.85**** | **0.73*** |
| Feo | **0.64*** | **0.66*** | 0.53 |
| Alp+0.5Fep | **0.81**** | **0.82**** | **0.70*** |
| Alp | **0.84**** | **0.85**** | **0.74*** |
| Fep | 0.55 | 0.58 | 0.54 |
| Alo-p+0.5Feo-p | 0.46 | 0.43 | 0.37 |
| Alo-p | 0.56 | 0.57 | 0.48 |
| Feo-p | 0.55 | 0.55 | 0.46 |
| Total C/Alp molar ratio | **-0.79**** | **-0.78**** | **-0.72*** |
| Cp | 0.39 | 0.42 | 0.23 |
| Cp/Alp molar ratio | **-0.79**** | **-0.75*** | **-0.74*** |

There were also considerable relations among Alp contents, total C/Alp molar ratio, and $qCO_2$_soc (Fig. 7). Soil Alp content and total C/Alp molar ratio showed a negative correlation at $p < 0.05$. The $qCO_2$_soc showed a positive correlation with

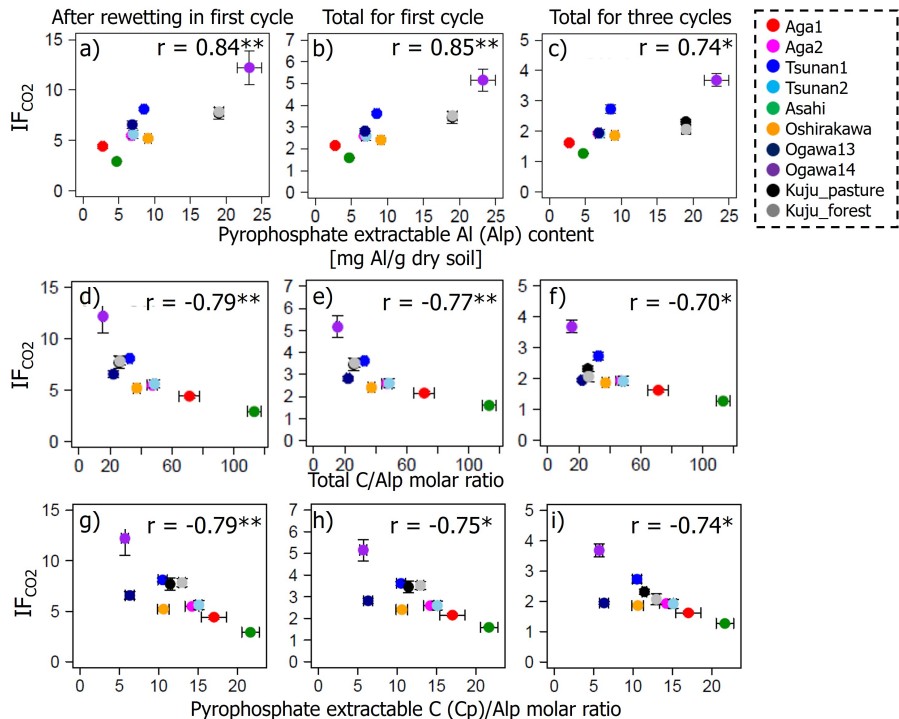

**Figure 5.** Relations between IF$_{CO2}$ and soil pyrophosphate-extractable Al (Alp) content (upper panels), total C to Alp molar ratio (middle panels), and pyrophosphate-extractable C (Cp) to Alp molar ratio (bottom panels). Significant correlation coefficients at $p < 0.01$ and $p < 0.05$ are indicated with single (*) and double asterisks (**), respectively.

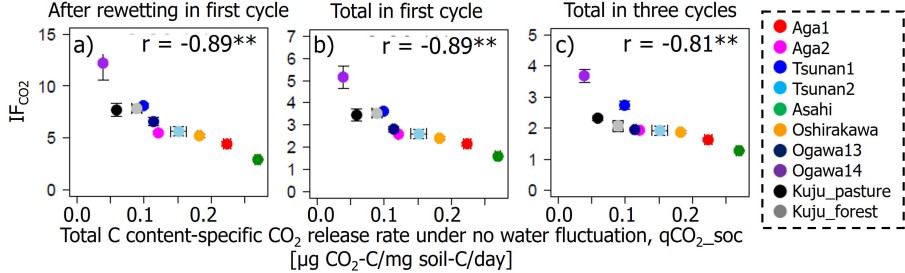

**Figure 6.** Relations between IF$_{CO2}$ and total C content-specific CO$_2$ release rate under constant moisture content (qCO$_2$_soc). Significant correlation coefficients at $p < 0.01$ and $p < 0.05$ are indicated with single (*) and double asterisks (**), respectively.

total C/Alp molar ratio and negative correlation with Alp content at $p < 0.01$. The Cp/Alp molar ratio also showed similar covariations with these variables, showing a strongly positive correlation with total C/Alp molar ratio ($r = 0.90$, $p < 0.01$; Fig. S3).

**Table 6.** Pearsons correlation coefficients between $IF_{CO2}$ and soil microbial properties

| Soil microbial property | After rewetting in first cycle | Total in first cycle | Total in three cycles |
|---|---|---|---|
| Total C content-specific $CO_2$ release rate under no water fluctuation, $qCO_2\_soc$ | **-0.87*** | **-0.86**** | **-0.77*** |
| $K_2SO_4$ extractable C | -0.25 | -0.21 | -0.33 |
| $K_2SO_4$ extractable N | -0.63 | -0.60 | -0.55 |
| $K_2SO_4$ extractable C/N | 0.15 | 0.18 | -0.01 |
| Microbial biomass C | -0.62 | -0.61 | -0.48 |
| Microbial biomass N | **-0.67*** | **-0.66*** | -0.52 |
| Microbial biomass C/N | 0.47 | 0.48 | -0.31 |

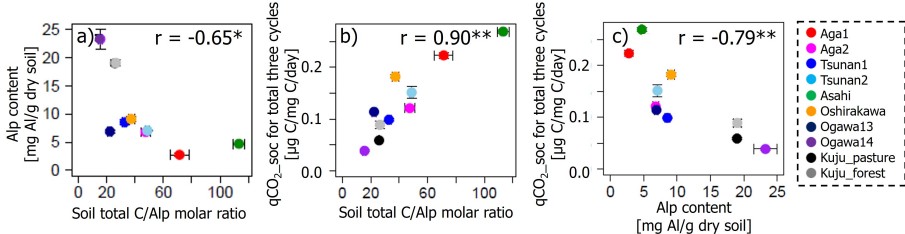

**Figure 7.** Relations among soil Alp contents, total C/Alp molar ratio, and $qCO_2\_soc$. Significant correlation coefficients at $p < 0.01$ and $p < 0.05$ are indicated with single (*) and double asterisks (**), respectively.

It should be noted that both microbial biomass C and N were significantly lower in soils incubated under DWC conditions than constant water content conditions ($p < 0.05$; Fig. 8). Microbial biomass C under DWC conditions was lower by  20.2% than those under constant water content conditions. Microbial biomass N under DWC conditions was  12.6% lower than under constant water content conditions. Nevertheless, there were no significant correlations between the differences in microbial biomass and the value of $IF_{CO2}$ ($p$  0.15; Fig. 9).

## 4   Discussion

An increase in $CO_2$ release due to DWCs was consistently observed across 10 forest and pastureland soils in Japan (Figs. 3 and 4). The comprehensive increases in $CO_2$ release by DWCs were different from the findings of a recent meta-analysis of studies showing no significant increases in $CO_2$ release under DWCs compared with constant water content with an equivalent mean water content during the period of interest (Zhang et al., 2020). Furthermore, our observations quantified the increase

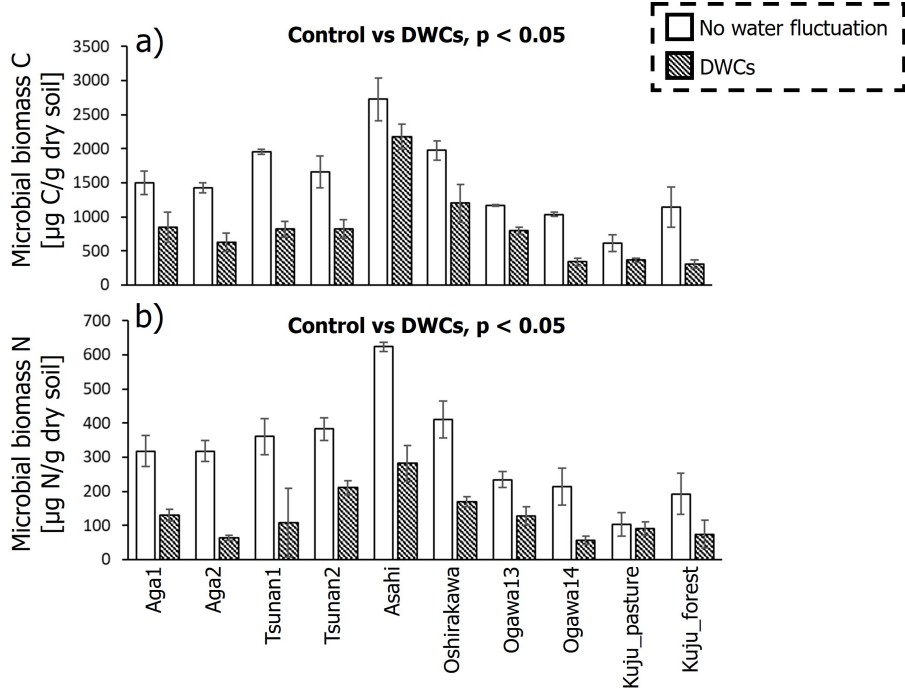

**Figure 8.** Comparisons of microbial biomass C (upper panel) and N (bottom panel) after incubation between DWC and constant water content conditions. Significant differences ($p < 0.05$ by pairwise $t$ test) in microbial biomass between two treatments are presented.

in $CO_2$ release due to three DWCs as $IF_{CO2}$ values of 1.3–3.7. The observed effect size of DWCs on the $CO_2$ release from soils was large given that even a 20% increase in $CO_2$ release from the worlds soils can exceed the annual $CO_2$ emission from anthropogenic processes (Friedlingstein et al., 2020).

Furthermore, analyzing the relations between $IF_{CO2}$ and fundamental soil properties showed a significantly positive correlation between $IF_{CO2}$ and soil Alp content (Table 5, Fig. 5). Of the organo-metal complexes measured as pyrophosphate-extractable metals, Fep is known to be sensitive to DWCs, especially in seasonally flooded forests and wetlands, likely due to their vulnerability to redox potential changes caused by alterations to the water regime (Lacroix et al., 2019; Chen et al., 2017, 2018; Chen and Thompson, 2018). Less is known about the vulnerability of Alp to DWCs. However, soil Alp content may be affected by DWCs through changes in soil acidity. A previous field survey conducted in Japanese forest and arable soils by Takahashi et al. (2006) showed that liming of non-allophanic Andisols increased soil pH and decreased Alp content. Miyazawa et al. (2013) verified this behavior by a laboratory incubation experiment for Andisols. Although we did not monitor pH during incubation in the present study, increases in pH after DWCs have been widely observed in upland agricultural (Wang et al., 2020; Meng et al., 2020), seasonally submerged paddy, and wetland soils (Majumdar et al., 2023). Therefore, the acidity mitigation by DWCs would destroy the organo-Al complexes, and increase microbially available C through the release of C protected by the organo-Al complexes or other soil elements, such as macro- and microaggregates, which are tightly bonded by

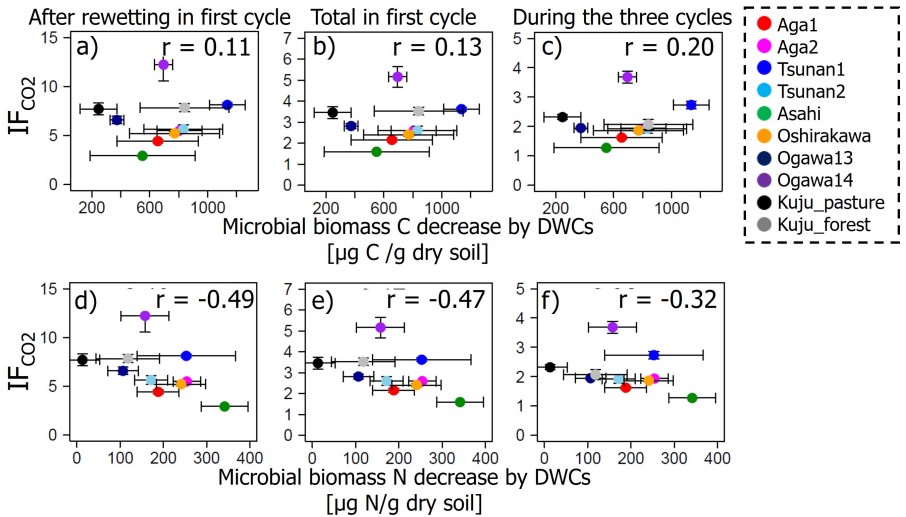

**Figure 9.** Relations between $IF_{CO2}$ and microbial biomass C or N decrease by DWCs. There were no statistically significant correlations ($p > 0.05$).

organo-Al complexes and thus physically protect organic C from microbial decomposition (Asano and Wagai, 2014; Takahashi and Dahlgren, 2016; Wagai et al., 2018).

Covariations among soil Alp contents, total C/Alp molar ratio, and qCO2_soc (Fig. 7) also support the state of soil Alp content as the primary predictor of variations in $IF_{CO2}$. Negative correlations of soil Alp contents with both total C/Alp molar ratio and qCO2_soc suggested that more Alp to total C (i.e., high total C/Alp values) strengthens binding between Alp and organic matter, resulting in resistance of organic matter to microbial decomposition under constant water conditions (i.e., low qCO2_soc values). These features among reactive Al and SOM dynamics would result in positive correlations of qCO2_soc

and total C/Alp molar ratio with $IF_{CO2}$ (Table 6, Figs. 5 and 6). However, such ability of Alp-rich soils to protect SOM from microbial decomposition likely does not persist under conditions of increased water fluctuations associated with DWCs, as suggested above. Given that amounts of pyrophosphate extractable C (Cp) in investigated soils (>19,000 ţg/g dry soil, Table 4), which represented C associated with the organo-metal complex, were substantially greater than observed $CO_2$ release increase by DWCs (620–1999 $\mu$g C g$^{-1}$ dry soil 84 days$^{-1}$), limited and specific C substrates in the organo-metal complex likely

associated with $CO_2$ increase by DWCs.

   We also found substantially lower microbial biomass in soils subjected to DWCs than constant water content conditions (Fig. 8), suggesting a decrease in microbial biomass through the destruction of microbial cells by DWCs (Kaiser et al., 2015; Marumoto et al., 1977, 1982; Marumoto, 1984; Nagano et al., 2023; Unger et al., 2010, 2012). The destruction of microbial cells is expected to release soluble organic matter available for microbes that have survived the DWC and to cause a marked

increase in $CO_2$ release after rewetting (Marumoto et al., 1977, 1982; Marumoto, 1984; Nagano et al., 2019; Unger et al., 2010, 2012). Nevertheless, the contribution of microbially derived substances to the increase in $CO_2$ release remained unclear

in the present study because of the lack of a significant correlation between $IF_{CO2}$ and the decrease in microbial biomass (Fig. 9). Whereas there was also no significant correlation between the decrease in microbial biomass and the increase in $CO_2$ release due to DWCs (Fig. S4), the decrease in microbial biomass C (246–1134 $\mu$g C g$^{-1}$ dry soil) was within the amount of increase in $CO_2$ release (620–1999 $\mu$g C g$^{-1}$ dry soil 84 days$^{-1}$). Therefore, the strict mechanisms of these carbon sources to $CO_2$ release increase, including the persistence and timing of their contribution, still require further works (Schimel, 2018; Barnard et al., 2020), considering the significant contribution of more than two carbon pools to the $CO_2$ release increase (Slessarev and Schimel, 2020; Warren and Manzoni, 2023), as well as carbon likely released by destructions of organo-metal complexes and microbial biomass by DWC as suggested in the present study.

It should be noted that shortcomings from unmeasured $CO_2$ release during the drying periods for DWCs treatments (i.e., Day 1 to Day 7 and Day 18 to Day 24 in each cycle) should be minor, while the linear changes in $CO_2$ release rate during the drying also assumed in other studies (e.g., Nagano et al. (2019); Zhang et al. (2020); Jin et al. (2023)). This is because we observed significant relationships between $IF_{CO2}$ and organo-Al complex also in $IF_{CO2}$ after the rewetting in addition to the $IF_{CO2}$ for total 1st cycle and three cycles (Table 5).

## 5   Conclusions

Through the present study, a comprehensive increase in $CO_2$ release by DWCs (i.e., 1.3–3.7-fold greater than $CO_2$ release under constant water conditions) was observed in Japanese forest and pastureland soils. These magnitudes of increase in $CO_2$ release were strongly correlated with soil Alp content, total C/Alp molar ratio, and total C content-specific $CO_2$ release rate under constant water conditions (i.e., qCO$_2$_soc), suggesting the possible vulnerability of SOM protection by organo-Al complexes against DWCs. A decrease in microbial biomass by DWCs was also suggested, whereas their relations with the increase in $CO_2$ release remains to be determined in future studies.

*Data availability.* The data that support the findings of this study are available from the corresponding author upon reasonable request.

*Author contributions.* YS and HN established the basic research design, conducted all data analyses, including software preparation, validation, and visualization, and wrote the original manuscript. SH, MA-A, JK, and HN conducted soil sampling. YS, SH, MA-A, JK, and HN contributed to the detailed research design, soil analysis, data validation, interpretation of the results, and editing of the manuscript. TY and YK provided essential support in analysis of soil properties and data. All authors contributed to editing the article and approved the submitted version.

*Competing interests.* The authors declare that they have no known competing financial interests or personal relations that could have appeared to influence the work reported in this paper.

*Acknowledgements.* This work was supported by the Japan Society for the Promotion of Science (JSPS) KAKENHI (grant numbers 21H02231, 21H05313, 22H05717). The authors thank Prof. Kouki Hikosaka and Dr. Hirofumi Kajino of Tohoku University for providing the vegetation information on the Oita forest site. Ms. Ayako Tamaki and Mr. Masahiro Otaki at Niigata University supported sample preparation and analysis during the experiment and text editing in preparing the manuscript. Dr. Rei Shibata at Niigata University and Dr. Masami Tsukahara, Mr. Kosuke Ito, and Mr. Tatsuki Tanaka at Niigata Prefectural Forest Research Institute helped in site selection and soil sampling in Niigata

Prefecture, Japan. Prof. Naoki Harada, Dr. Kazuki Suzuki, and Dr. Asiloglu Rasit of Niigata University and Dr. Hiroyuki Sase and Dr. Rieko Urakawa of Asia Center for Air Pollution Research (ACAP) provided valuable preliminary discussion in writing the manuscript. The authors also thank Ms. Misuzu Kaminaga, Ms. Kikuko Yoshigaki, Ms. Makiko Ishihara, and Ms. Kazumi Matsumura at JAEA for support with laboratory work.

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
