# Peer review of "Comprehensive increase in CO2 release by drying-rewetting cycles among Japanese forests and pastureland soils and exploring predictors of increasing magnitude"

_EGUsphere, 2024_

## Author Comment (AC1)

Thank you very much for evaluating our manuscript positively. Followings are our response to your concerns. We hope our reply and suggested revisions will satisfy your concerns.

**1) About different water contents among soils:**

In our study, we consider that the soil water content at the soil sampling reflected the ability of soil to hold the water and thus the usual water contents in the field because the soil water content showed significantly positive correlations with WHC (r = 0.87, p<0.01). Nevertheless, none of the soil water content at the soil sampling and the water holding capacity (WHC) showed a significant relation with the increasing factor of $CO_2$ release by DWC ($IF_{CO2}$) not only as a linear correlation but also as a nonlinear relation, which was examined visually (Table 4; also see the figure below). These facts from the obtained data support us in stating that the variations in IFCO2 were significantly associated with soil metal-humus complexes and soil microbiology rather than different soil water content among soils (Tables 4 and 5, Fig 5). We will add the description of the positive correlation between the soil water content and WHC to the material and method section of the soil sampling (L80 to L88), especially to clarify the relation between the soil water content and the WHC.

[Figure]

Figure. The relations between $IF_{CO2}$ and soil water content at the soil sampling in the field. The relations against $IF_{CO2}$ after rewetting in first cycle (a), as the total for first cycle (b), and as the total for three cycles (c). There was no significant correlation between the two variables.

**2) About aerobic conditions for the incubation experiment:**

The situation of the incubation experiment allows us to consider that the incubated soils have been aerobic even after the rewetting to increase the water content by twice the WHC. The primary evidence supporting this is that the CO2 concentrations in our experiment never overwhelmed 1%, thus the oxygen concentrations in the incubation jar have likely never decreased below 19% or lower. Also, a sufficiently large volume of our incubation jar (1.0L) against contained soil amounts (i.e., 5.31–10.63 g) and added water contents in the rewetting (i.e., ca. 6 to 7 mL) support the state of aerobic condition during the incubation. We will add this

description to the material and method section of the incubation experiment (L110 to L131) to help the reader understand the aerobic conditions during the incubation.

**3) About the mechanisms for large increases in $CO_2$ release after rewetting:**

In our study, we suggested the two carbon sources that contributed to the increase in $CO_2$ release by DWC. One is the destruction of microbial cells by rewetting, which is more well-known than another factor, such as the release of carbon associated with the organo-metal complex. As you pointed out, microbial biomass carbon only accounts for a minimal fraction of SOC (i.e., ca. 1%). However, when considering the quantitative relationship between the amount of $CO_2$ increased by the dry-wet cycle (ca. 620 to 2,000 µg C/g dry soil) and the decreased amount of microbial biomass carbon (ca. 250 to 1,100 µg C/g dry soil), the microbial biomass decreases could contribute up to 64% of the $CO_2$ release increase during the 84-days incubation. In addition to the microbial biomass carbon, investigated soils contained 19,000 µg/g dry soil or more of pyrophosphate extractable-carbon (Cp; Table 3), which partially represented carbon associated with the organo-metal complex. Moreover, there were significantly positive correlations between $IF_{CO2}$ and such organo-metal complex contents measured as pyrophosphate extractable aluminum (Alp) and iron (Fep) (Table 4). Especially, Alp showed consistent relations with $IF_{CO2}$ in the present study (Fig. 5), suggesting Alp as the primary predicting factor for the $IF_{CO2}$ variations. Thus, the carbon associated with the organo-metal complex, in addition to the destroyed microbial biomass, was suggested as the likely primary carbon sources contributing to the $CO_2$ release increase by DWC. Nevertheless, the strict mechanisms of these carbon sources to $CO_2$ release increase, including the persistence and timing of their contribution, still require further works (Schimel, 2018; Barnard et al., 2020), considering the significant contribution of more than two carbon pools to the $CO_2$ release increase (Slessarev and Schimel, 2020; Warren and Manzoni, 2023). To clarify these points, we will refine our sentences in the discussion section, especially from L201 to L236 and the conclusion (L238-L244), adding the references mentioned above (i.e., Schimel, 2018; Barnard et al., 2020; Slessarev and Schimel, 2020; Warren and Manzoni, 2023).

We will also add the identification number for each for the sub-panel in figures with more than one sub-panels (i.e., Fig 2-9).

References to be cited to the revised manuscript:
- Barnard, R.L., Blazewicz, S.J., Firestone, M.K., 2020. Rewetting of soil: Revisiting the origin of soil CO2 emissions. Soil Biology and Biochemistry 147, 107819. doi:10.1016/J.SOILBIO.2020.107819

- Schimel, J.P., 2018. Life in Dry Soils: Effects of Drought on Soil Microbial Communities and Processes. Annual Review of Ecology, Evolution, and Systematics 49, 409–432. doi:10.1146/annurev-ecolsys-110617-062614
- Slessarev, E.W., Schimel, J.P., 2020. Partitioning sources of $CO_2$ emission after soil wetting using high-resolution observations and minimal models. Soil Biology and Biochemistry 143, 107753. doi:10.1016/j.soilbio.2020.107753
- Warren, C.R., Manzoni, S., 2023. When dry soil is re-wet, trehalose is respired instead of supporting microbial growth. Soil Biology and Biochemistry 184, 109121. doi:10.1016/J.SOILBIO.2023.109121

---

## Author Response (AR1)

Dear RC1,

Thank you very much for evaluating our manuscript positively. Followings are our responses to your valuable comments to the manuscript. The places where revisions were made in the manuscript were highlighted with yellow in this response file. According to the revisions, we made and added new Table 3 for soil carbon (C) and (N) properties. Therefore old Tables 3, 4, and 5 should be Tables 4, 5, and 6, respectively, in the revised manuscript.

*RC1: 'Comment on egusphere-2024-419', Anonymous Referee #1, 11 May 2024*
*The authors investigated the effects of drying-rewetting cycles (DWCs) on soil CO2 release and explored the controlling environmental and soil predictors for variations in the effects based on incubation experiments using 10 Japanese forests and pastureland soils. The topic of this study is interesting and important, and fits well the scope of this journal. The manuscript is well-structured and presented clearly. Nonetheless, I still have two major concerns on the methods and results of this study:*

➢ Followings are our response to your concerns, as well as we provided with the same context in the interactive discussion. We believe our reply and applied revisions satisfy your concerns.

*To give a fair comparison of DWC-induced change in CO2 release rates between different soil samples, the constant soil water content and DWC for each soil samples should be same. However, the soil water contents for different soil samples (Fig. 3) are very different in this study. This might disturb the results of this study. The difference in responses of CO2 release rate to DWC might just because the different constant soil water content, rather than the environmental and soil preditors. In addition, the authors claimed that the soils were incubated aerobically. Yet from Fig. 3, we can find that the soil water contents during the rewetting period could be 1 to 2 times of the WHC. To my understanding, the soil will be in anaerobic environment when the actual soil water content exceeds WHC.*

➢ **About different water contents among soils:** In our study, we consider that the soil water content at the soil sampling reflected the ability of soil to hold the water and thus the usual water contents in the field because the soil water content showed significantly positive correlations with WHC ($r = 0.87$, $p<0.01$). Therefore, $CO_2$ release rate for constant moisture conditions in the present study should represent the release rate under the usual field moisture conditions of each soil. Nevertheless, none of the soil water content at the soil sampling and the water holding capacity (WHC) showed a significant relation with the increasing factor of $CO_2$ release by DWC ($IF_{CO2}$) not only as a linear correlation but also as a nonlinear relation, which was examined visually (Table 5; also see the figure below). These facts from the obtained data support us in stating that the variations in $IF_{CO2}$ were significantly associated

with soil metal-humus complexes and soil microbiology rather than different soil water content among soils (Tables 5 and 6, Fig 5). We added the descriptions of the positive correlation between the soil water content and WHC to the material and method section of the soil sampling (L95 to L99), especially to clarify the relation between the soil water content and the WHC. Then, we also added no significant relationship between $IF_{CO2}$ and any soil water content to the result section for exploring predictors of $IF_{CO2}$ (L199 to L201).

[Figure]

Figure. The relations between $IF_{CO2}$ and soil water content at the soil sampling in the field. The relations against $IF_{CO2}$ after rewetting in first cycle (a), as the total for first cycle (b), and as the total for three cycles (c). There was no significant correlation between the two variables.

> **About aerobic conditions for the incubation experiment:** The situation of the incubation experiment allows us to consider that the incubated soils have been aerobic even after the rewetting to increase the water content by twice the WHC. The primary evidence supporting this is that the $CO_2$ concentrations in our experiment never overwhelmed 1%, thus the oxygen concentrations in the incubation jar have likely never decreased below 19% or lower. Also, a sufficiently large volume of our incubation jar (1.0L) against contained soil amounts (i.e., 5.31–10.63 g) and added water contents in the rewetting (i.e., ca. 6 to 7 mL) support the state of aerobic condition during the incubation. We added this description to the material and method section of the incubation experiment (L132 to L137) to help the reader understand the aerobic conditions during the incubation.

*The explanation on the higher CO2 release rate under DWC than that under constant soil water content might be not that convinced. It is surprising to see that the CO2 release rate is very high during the rewetting period when the soil is in anaerobic environment. The authors argue that the destruction of microbial cells is expected to release soluble organic matter available for microbes that have survived the DWC and to cause a marked increase in CO2 release after rewetting. But will this mechanism continue for a long time? Microbial biomass*

*only accounts for very limited fraction of SOC. Even the destruction of microbial cell can contribute to the release of soluble organic matters, I am doubt if this contribution can result in such a significant increase in the CO2 release rate, in particular under anaerobic environment. In addition, the microbial biomass would declined quickly during the DWC experiment, will the higher CO2 release rate continue for a longer time?*

➢ In our study, we suggested the two carbon sources that contributed to the increase in $CO_2$ release by DWC. One is the destruction of microbial cells by rewetting, which is more well-known than another factor, such as the release of carbon associated with the organo-metal complex. As you pointed out, microbial biomass carbon only accounts for a minimal fraction of SOC (i.e., ca. 1%). However, when considering the quantitative relationship between the amount of $CO_2$ increased by the dry-wet cycle (620 to 1,999 µg C/g dry soil) and the decreased amount of microbial biomass carbon (246 to 1,134 µg C/g dry soil), the microbial biomass decreases could contribute up to 64% of the $CO_2$ release increase during the 84-days incubation. In addition to the microbial biomass carbon, investigated soils contained 19,000 µg/g dry soil or more of pyrophosphate extractable-carbon (Cp; Table 4), which partially represented carbon associated with the organo-metal complex. Moreover, there were significantly positive correlations between $IF_{CO2}$ and such organo-metal complex contents measured as pyrophosphate extractable aluminum (Alp) and iron (Fep) (Table 5). Especially, Alp showed consistent relations with $IF_{CO2}$ in the present study (Fig. 5), suggesting Alp as the primary predicting factor for the $IF_{CO2}$ variations. Thus, the carbon associated with the organo-metal complex, in addition to the destroyed microbial biomass, was suggested as the likely primary carbon sources contributing to the $CO_2$ release increase by DWC. Nevertheless, the strict mechanisms of these carbon sources to $CO_2$ release increase, including the persistence and timing of their contribution, still require further works (Schimel, 2018; Barnard et al., 2020), considering the significant contribution of more than two carbon pools to the $CO_2$ release increase (Slessarev and Schimel, 2020; Warren and Manzoni, 2023). To clarify these points, we refined our sentences in the discussion section, especially from L257 to L260 and L270 to L274, adding the references mentioned above (i.e., Schimel, 2018; Barnard et al., 2020; Slessarev and Schimel, 2020; Warren and Manzoni, 2023).

*Please add the identification number for each for the sub-panel in figures with more than one sub-panels.*

➢ We added the identification number for each for the sub-panel in figures with more than one sub-panels (i.e., Fig 2-9).

References additionally cited to the revised manuscript:

- Barnard, R.L., Blazewicz, S.J., Firestone, M.K., 2020. Rewetting of soil: Revisiting the origin of soil $CO_2$ emissions. Soil Biology and Biochemistry 147, 107819. doi:10.1016/J.SOILBIO.2020.107819

- Schimel, J.P., 2018. Life in Dry Soils: Effects of Drought on Soil Microbial Communities and Processes. Annual Review of Ecology, Evolution, and Systematics 49, 409–432. doi:10.1146/annurev-ecolsys-110617-062614

- Slessarev, E.W., Schimel, J.P., 2020. Partitioning sources of $CO_2$ emission after soil wetting using high-resolution observations and minimal models. Soil Biology and Biochemistry 143, 107753. doi:10.1016/j.soilbio.2020.107753

- Warren, C.R., Manzoni, S., 2023. When dry soil is re-wet, trehalose is respired instead of supporting microbial growth. Soil Biology and Biochemistry 184, 109121. doi:10.1016/J.SOILBIO.2023.109121

Dear RC2,

Thank you very much for your kind evaluations of our manuscript. Followings are our responses to your valuable comments to the manuscript. The places where revisions were made in the manuscript were highlighted with yellow in this response file. According to the revisions, we added new Table 3 for soil carbon (C) and (N) properties. Therefore old Table 3, 4, and 5 should be Tables 4, 5, and 6, respectively, in the revised manuscript.

*RC2: 'Comment on egusphere-2024-419', Anonymous Referee #2, 28 Jun 2024*
*The current study deals with a relevant and timely topic: How would moisture fluctuation impact organic matter mineralization in volcanic soils vs. under constant moisture regime. Some intriguing result comes out: the contrast in CO2 emission from soil depends on the pyrophosphate or NH4-oxalate extracted Al (and Fe) level. The authors provided a first interpretation of the found results and also tried to link observed higher soil CO2 emission with lower microbial biomass C under fluctuating vs. under constant moisture. While mostly the experiments seem to have been properly carried out, it is impossible to appreciate this with no details provided onto how soil moisture was monitored during the experiment.*

➤ First, I want to clarify our measurement of soil water content during the incubation period. Yes, we have periodically measured soil water content during the incubation, even in the drying stage under the DWC treatment for Day 1 to Day 7 and Day 18 to Day 24. For each drying stage, we conducted measurements of soil water content once to twice. The measurements were performed by weighing those soils. Based on these data, we confirmed that the mean soil water content during DWC incubation was equal to that during constant moisture incubation. Figure 3 (bottom panels) shows these data on soil water content. We have written some explanations for measuring soil water content at the Materials and Methods, but this was somewhat unclear. In the revision, we provided more explicit explanations for measuring soil water content during the incubation, as described above (the revisions appear at L155-L159).

➤ Additionally, under the constant water treatment, we surrounded the small vial with 20 mL of water within the incubation jar to prevent the soil from drying. Sorry for lacking this explanation. We added this explanation to L140-L142 in the Materials and Methods.

➤ We believe our explanations on measuring soil water content during the incubation solve some of your primary concerns in our study.

*Also, the interpretation of the findings does not really go in depth and we may only guess about the nature of the dependence of Birch-effect magnitude on Al and Fe contents in these soils. The main cause of this is probably that some essential soil information is missing, viz. soil*

*texture, soil moisture retention characteristics and any appreciation of the soil organic matter quality; Without these, it may well be that soil Al and Fe contents covaried with soil texture which should have a big impact on the moisture fluctuation under the imposed drying and rewetting. It is well known that it is notoriously hard to just even quantify soil texture in volcanic soils precisely owing to the very strong binding of soil particles by Al and Fe. Content of Fe and Al (hydr)oxides thus likely strongly impacts the soil moisture retention characteristic of such volcanic soils. The resulting effect is that magnitude of drying and rewetting might have been very different between the 10 investigated soils. In my view this paper is now to be resubmitted after soil textural and moisture retention characteristic have been provided and accounted for in the interpretation. Lastly, since mostly forest soils were included, a substantial part of the SOM may be present under the form of POM – it thus seems relevant enough to also carry out a limited soil fractionation and include POM and MAOM proportions as potential predictor variables of the IFCO2.*

➢ Thank you for your suggestions. We additionally measured soil textures as particle size distributions, i.e., relative compositions of clay, silt, and sand-sized particles (data are shown in Table 2). Then, we found that the amounts of clay and sand-sized particles showed significant correlations with $IF_{CO2}$ after rewetting in 1st cycle ($p < 0.05$, $r = -0.66$ for clay and 0.71 for sand particles). However, those correlations between the particle contents and $IF_{CO2}$ were insignificant for $IF_{CO2}$ for a total of three cycles. Nevertheless, as described in the previous manuscript, pyrophosphate-extractable Al (Alp) content showed significant correlations with $IF_{CO2}$ for all incubation stages ($r = 0.84$ to 0.74 with $p < 0.05$). These results support our argument that pyrophosphate-extractable Al (Alp) content is likely the primarily important factor for the magnitude of $CO_2$ release increase by DWC. We added these results and descriptions to the revised manuscript (L108 to L110 for the description of methodology of soil texture and L205 to L209 and Table 5 for the result).

➢ Regarding water retention, we consider the water-holding capacity (WHC) of soils. As presented in Table 2, we have measured WHC in addition to soil water content as soil properties used for the experiment. Nevertheless, none of the soil water content at the soil sampling and the WHC showed a significant relation with $IF_{CO2}$ not only as a linear correlation but also as a nonlinear relation, which was examined visually (Table 5; please also see the figure in our reply to RC1), while WHC and soil water content showed a significantly positive correlation to each other ($r = 0.87$, $p<0.01$; thus, soil water content at the soil sampling also reflected the ability of soil to hold the water and the usual water contents in the field, as described in L95 to L98). These facts from the obtained data support us in stating that the variations in $IF_{CO2}$ were significantly associated with soil metal-humus complexes and soil microbiology rather than different WHC among soils (Tables 5 and 6,

Fig 5). We refined our sentences for the results of exploring predictors of $IF_{CO2}$ in the manuscript (L194 to L209), adding the positive correlation between the soil water content and WHC to the material and method section of the soil sampling (L95 to L98).

- For the soil organic matter quality, we can consider the C/N ratio of $K_2SO_4$ extractable organic matter in addition to that ratio of total organic matter. Their correlations with $IF_{CO2}$ were statistically insignificant, as presented in Tables 5 and 6. Because POM is often referred to as the free light density fraction (fLF; Leuthold et al., 2023), we measured fLF contents especially for Kuju forest and grassland soils and two Ogawa forest soils by density fractionation with sodium polytungstate solutions having 2.0 $cm^{-3}$ as the threshold density (data shown in Table 3). Whereases fLF contents differed almost two-fold between Kuju forest and grassland soils (Table 3), $IF_{CO2}$ values differ only 12% between these two soils (Fig. 4). Moreover, two Ogawa forests soils differed almost two-fold in $IF_{CO2}$ values (Fig. 4) while they had similar fLF contents (Table 3). Given those facts for $IF_{CO2}$ and fLF contents of soils, fLF and thus POM likely had minor effects on observed variations in $IF_{CO2}$ among soils in the present study. We added these descriptions for no significant relationship of SOM quality with $IF_{CO2}$ to L201 to L205. Descriptions for methodology of measuring fLF was added to L1119 to L122.

- Those explanations will satisfy your concerns about insufficient soil information and evaluation for soil texture, soil moisture retention characteristics, and soil organic matter quality.

*More specific comments:*
*The introduction section starts off well, but from L44 till 56 its added value becomes limited: a listing of several studies that have tried to quantify the differences in soil CO2 emissions at constant and variable moisture level is not enough. In this introduction at least some brief overview needs to follow on current explanations for the generally observed higher overall CO2 efflux with variable vs. constant moisture level. Of particular relevance is to see if there were already any previous ideas on the fate of Fe/Al associated OM under variable vs; constant moisture? This then needs to lead towards formulation of a research hypothesis specifically connected to the potential Birch effect size for SOM in the Japanese soils, particularly considering the abundant presence of short-range ordered Fe/Al and its role in stabilizing OM. Without such parts it is not clear what the added value would be of this study.*

- Because our first primary purpose is to clarify the overall trend of DWC effect on soil $CO_2$ release under the comparison between DWC and constant moisture conditions, which have the same mean water content during the incubation, we less mentioned the proposed mechanism for $CO_2$ release increases under DWC in the introduction section. Roughly three

mechanisms are proposed (Schimel, 2018; Barnard et al 2020): (i) increase in available carbon source via the releases of cellular metabolites from microbial cells destroyed by rewetting after the strong drought, (ii) increase in available carbon source by the releases of carbon from macroaggregates destroyed by repeated DWC, and (iii) changes in the microbial communities in response to transient moisture conditions. Whereas the DWC-induced destructions of macroaggregates might be related to changes in association between mineral/metal and organic matters, we cannot find any literature which specifically mentioned the organo-Al complexes, which was found as to be the primary predictor for $IF_{CO_2}$ in the present our study. We added the brief description of currently proposed mechanisms for $CO_2$ release increase under DWC to the introduction section, especially after the paragraph to describe the current knowledge about the trend of DWC effects on soil $CO_2$ release in L57 to L62.

*Surprisingly, no motivation is given as to why the experiment was set up with these 10 particular soils. It is also difficult to compare these soils as some essential information to interpret the results is missing: soil particle size distribution, soil water retention characteristics and basic information on the soil organic matter quality. Especially in the forested sites it could be that much of the SOM occurs as particulate organic matter and that could form a contrast to the grassland sites. But without some basic soil fractionation data we cannot appreciate this.*

➢ This was that these soils were variously affected by volcanic ash during their pedogenesis, and therefore include several Andisols, which are known to have a high SOM storage capacity, likely due to the protection of SOM from microbial decomposition by enrichment of reactive minerals and metals in these soils. Our previous study showed the significant increase in $CO_2$ release by DWC for two Japanese forest soils. Especially, a volcanic ash soil showed a substantially large increase. This was why we used those 10 Japanese soils differently affected by volcanic ash. Whereas we have presented those information in the previous version manuscript, this might be not clear. We thoroughly refined those sentences in the revised manuscript, to present clear description of our motivation using 10 Japanese soils (L63 to L73).

*It is striking that soil moisture content was apparently not monitored during the soil incubation experiment – unless it was (?) -   But at least that was not described anywhere in the M&M. Given that only very small cores were used (containing but 8-10g) inside 1L jars it seems probable that in the constant soil moisture treatments actually soil dried out. But no data is provided to check this. The presented course of soil water during the DWC treatments with linear drying of soil and constant levels in between drying events seems unrealistic and should*

*have been replaced by actually recorded moisture. Because of this lack of moisture data we cannot be certain that the observed correlations between the IFCO2 and contents of Alo and Feo and Alp were not largely or in part indirect. Is it not conceivable that in soil having more OM and more pedogenic oxides the fluctuation of moisture differs to soils with less? Variation in these properties likely also causes a contrast in the soil moisture retention curve of these soils and that in turn will directly impact the magnitude of the imposed drying to actual soil moisture fluctuation.*

➢ We did the monitoring soil moisture during the incubation, as we described above (L95 to L98).

*Another major shortcoming is the non-continuous follow-up of soil CO2 emission: 1° CO2 emission from the constant moisture treatments was apparently only measured in the first 29 days; 2° Moreover no CO2 emissions were measured during the drying stages. Without these data, can we really compare emissions at constant and fluctuating moisture properly?*

➢ Sorry for confusing you. Our $CO_2$ measurement for constant moisture condition were conducted periodically during the 84-day incubation. Namely, in the incubations with the constant moisture condition, the $CO_2$ release rates were measured for Day 1 to Day 12, Day 13 to Day 28, Day 29 to Day 40, Day 41 to Day 56, Day 57 to Day 68, and Day 69 to Day 84, as shown in Figure 3. We revise sentences in L152 to L155 to describe the measurement of $CO_2$ release for the constant moisture conditions. Because we observed significant absorption of $CO_2$ by silica gels in preliminary experiments, we did not conduct the measurement of $CO_2$ release during the drying stages. Other previous studies also assumed the linear changes in $CO_2$ release rate during the drying. Furthermore, significant relationships between $IF_{CO2}$ and organo-Al complex was also observed in $IF_{CO2}$ after the rewetting in addition to the $IF_{CO2}$ for total 1st cycle and three cycles. Therefore, shortcomings from unmeasured $CO_2$ release during the drying periods should be minor as the uncertainty in our main findings in the present study. We added those description for Discussion section (L275 to L279).

*Details:*

*L44 "…in comparison with the medium level of constant moisture content" is not clear, what is meant by medium level here?*

➢ The medium level means constant moisture content equivalent to the mean water content during DWC incubation. For example, medium level should be 50% (w/w) of soil water content in the case with DWCs having 5% as the minimum water content and 95% as the maximum water content during the 28-day incubation having two of 7-day drying period,

each of 5-day periods of driest and wettest stages, and a 4-day period with moderately wetted stage (i.e., 50% of water content). To clarify this meaning of "the medium level", we refined our sentence in L43 to L46.

*L46 "Another 29 data were calculated…" sounds awkward and furthermore with this sentence you are not bringing any message: what was now the outcome of this comparison?*

➢ The most important thing is 29 data of 28 data originates from just calculations not from actual measurements. To clarify this point, we thoroughly refined our sentence in L46 to L48.

*It is not clear really what is intended by "soil carbon content-specific CO2 release rate under continuous constant moisture conditions (qCO2_soc)" – requires further clarification*

➢ The qCO2_soc should be an index for soil total C normalized availability of carbon substrate for soil microbes. Thus, using qCO2_soc, we can consider whether the microbially available carbon substrate in interested soil was much more than that in other soils. Then, we would also be able to evaluate the linkage of qCO2_soc with $IF_{CO_2}$ as discussed in L250 to L257. To clarify the meaning of qCO2_soc, we refined our sentences in L213 to L215.

*Fig 2 would be useful to indicate the point in time what interval this 'after rewetting in first cycle' now precisely ended*

➢ Thanks. We added drawings to clarify that to Fig. 2.

*L94 to measure field capacity, likely also soil was allowed to leach out after its saturation? But that is not well described here.*

➢ WHC (water holding capacity) is not equal to field water capacity. We measured WHC with the Hilgard method (Mabuhay et al. 2003, Ahn et al. 2008). Here, water contents when soil is completely saturated should be equal to zero pF value (0 kPa) as soil water potential. To clarify the methodology of WHC, we refined our sentence in L104 to L107.

*L194 "Especially, the importance of Alp for variations in IFCO2… " this link between Alp content comes in too early and is best omitted from this start of the discussion section.*

➢ OK, we now deleted these sentences (L235 to L236).

*L196 Better not directly make a leap towards podzols, safer to just restrict the interpretation to volcanic soils.*

➢ We also deleted these sentences (L235 to L236).

***L 234 the link to CH4 and N2O emission is best not made.***

➢ OK, we now deleted these sentences (L270).

***L238 A strange starting sentence 'insight in the precise quantification' needs to be revised***

➢ We understood and deleted these sentences (L281).